# MAGAN: ALIGNING BIOLOGICAL MANIFOLDS

**Matthew Amodio**
Department of Computer Science
Yale University
matthew.amodio@yale.edu

**Smita Krishnaswamy**
Department of Genetics
Department of Computer Science
Yale University
smita.krishnaswamy@yale.edu

## ABSTRACT

It is increasingly common in many types of natural and physical systems (especially biological systems) to have different types of measurements performed on the same underlying system. In such settings, it is important to align the manifolds arising from each measurement in order to integrate such data and gain an improved picture of the system. We tackle this problem using generative adversarial networks (GANs). Recently, GANs have been utilized to try to find correspondences between sets of samples. However, these GANs are not explicitly designed for proper alignment of manifolds. We present a new GAN called the Manifold-Aligning GAN (MAGAN) that aligns two manifolds such that related points in each measurement space are aligned together. We demonstrate applications of MAGAN in single-cell biology in integrating two different measurement types together. In our demonstrated examples, cells from the same tissue are measured with both genomic (single-cell RNA-sequencing) and proteomic (mass cytometry) technologies. We show that the MAGAN successfully aligns them such that known correlations between measured markers are improved compared to other recently proposed models.

## 1 INTRODUCTION

We commonly face the situation of having samples from a pair of related domains and want to ask the natural question of how samples from one relate to samples from the other. Our motivational system for this is two types of measurements on cells sampled from the same population in a biological system. It is important for the discovery of new biology to integrate these datasets, which are often generated at great cost and expense. However, a fundamental challenge is that there are exponentially many possible relationships that could exist between the two domains of measurement and the system must learn a logical way to map between them.

The first approaches for teaching neural networks to learn these relationships required supervised paired examples from each domain, an impractical demand for many applications Isola et al. (2016). Recently, there have been attempts at performing the same task without the supervision of paired data Zhu et al. (2017); Yi et al. (2017); Kim et al. (2017). Like these previous models, the MAGAN learns to map between distinct domains from unsupervised, unpaired data without pretraining. It can take a point in the first domain and generate a point that is indistinguishable from points in the other domain. However, unlike previous models, the MAGAN learns the most *coherent* mapping, rather than an arbitrary one. The MAGAN will not just take a point in the first domain and generate any point from the second domain, but it will generate the most closely related one. This is achieved by aligning, rather than superimposing, the manifolds of the two domains.

The high-dimensional inputs that are typical for neural network applications can typically be modeled very well with a lower-dimensional manifold Domingos (2012). Much work has framed the generation problem of GANs as sampling points from this manifold Park et al. (2017); Zhu et al. (2016). Here, each domain lies on a manifold and we want to find an alignment between them.

We first consider an example of the difference between superimposing and aligning manifolds on image domains. Earlier work Kim et al. (2017) has demonstrated that an image of an object in the first domain can be mapped to an object of another object in the second domain while preserving the

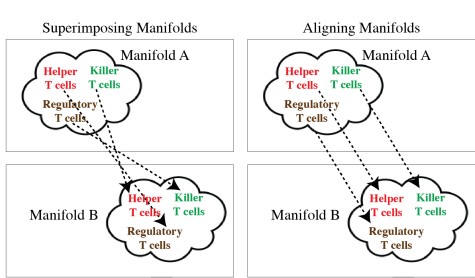

Figure 1: There are exponentially many mappings that superimpose the two manifolds, fooling a GAN's discriminator. By aligning the manifolds, we maintain pointwise correspondences.

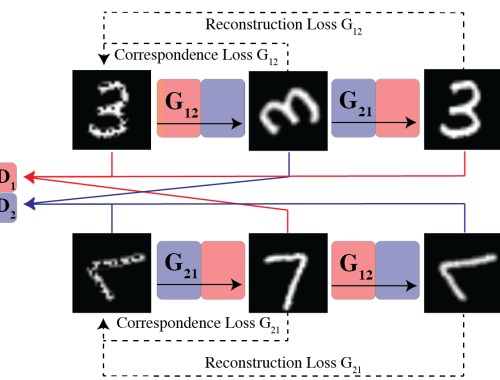

Figure 2: The MAGAN architecture with two generators, two discriminators, reconstruction loss, and correspondence loss. Domain 1 comprises upright images of 3's and 7's, Domain 2 comprises rotated images of 3's and 7's.

orientation with respect to the picture frame. However, in those cases, the orientation axis can be completely reversed. An image in the first domain facing $30°$ maps to an image in the second domain facing $150°$, and vice versa. The mappings successfully fool the discriminator in each domain at the level of an entire batch (the manifolds are superimposed), but there are other mappings that also fool the discriminator that preserve the individual pointwise structure of the original domain. Namely, the optimal alignment would map first domain images at $30°$ to second domain images also at $30°$. Without aligning the manifolds, only random initialization will determine which superimposition is learned on any particular attempt.

While the preference for the logical mappings of manifold alignment over the arbitrary ones of manifold superimposition is of general interest to all domains, we present multiple applications in single-cell biological analysis where it is essential. We propose the novel concept of using adversarial neural networks for alignment of manifolds arising from different biological experimental data measurement types.

Single-cell biological experiments create many situations where manifold alignment problems are of interest. New technologies allow for measurements to be made at the granularity of each cell, rather than older technologies which could only acquire aggregate summary statistics for whole populations of cells. While these instruments allow us to discover biological phenomena that were not apparent before, it is a challenge to integrate and analyze this information in a unified fashion for biological discovery. Further, even for the same technology, experiments run on different days or in different batches can show variations even on the same populations, possibly due to calibration differences. In such cases even replicate experiments need alignment before comparison. Two such technologies that we examine are single-cell RNA sequencing which measures cells in thousands of gene (mRNA) dimensions and mass cytometry which measures protein abundances in several dozen dimensions Bendall et al. (2012); Klein et al. (2015).

In all of these examples we have two data manifolds with a latent physical cell being measured analogously in each manifold. In some applications it might be adequate to simply superimpose these manifolds in any way. In many applications though, including the ones demonstrated here, we would like to be able to align them such that the two representations of each latent cell are aligned. The MAGAN presented here improves upon neural models for manifold alignment by finding the mapping between the manifolds (*correspondence*) that models these latent points by penalizing differences in each point's representation in the two manifolds.

We summarize the contributions of this paper as follows:

1. The introduction of a novel GAN architecture that aligns rather than superimposes manifolds to find relationships between points in two distinct domains

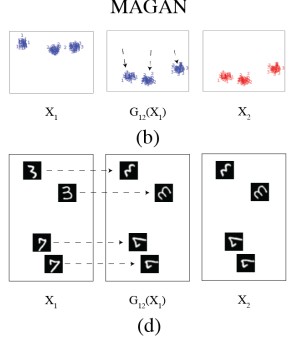

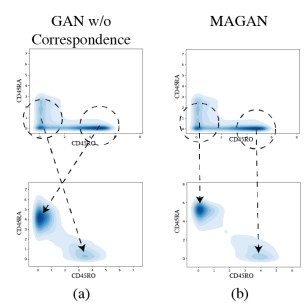

Figure 3: Both models superimpose the manifolds, (the first domain ($X_1$) is mapped to the second domain ($X_2$) such that after mapping ($G_{12}(X_1)$) matches $X_2$). Without the correspondence loss, this mapping is arbitrary and thus the relationships found vary. With the correspondence loss, the relationships found are coherent. This is confirmed with (a) a GAN without correspondence loss on artificial data (b) MAGAN on artificial data (c) a GAN without correspondence loss on MNIST and (d) MAGAN on MNIST.

Figure 4: (a) Without correspondence loss, the GAN corrects the batch effect but subpopulations are reversed. (b) The MAGAN still corrects the batch effect and subpopulations are preserved.

2. The demonstration of novel applications made possible by the new architecture in the analysis of single-cell biological data

We demonstrate the MAGAN's performance on artificial data and the standard MNIST dataset. We then apply it to three real-world biological applications: mapping between two replicate cytometry domains, mapping between two different cytometry domains, and mapping between one cytometry domain and a single-cell RNA sequencing domain.

# 2 EXPERIMENTS

We demonstrate that the MAGAN with its correspondence loss preserves crucial information that is lost with a mapping from a GAN without the correspondence loss. In Figure 3a and 3b, there are two constructed domains each composed of 3 subpopulations of Gaussians. Without the correspondence loss, subpopulations are arbitrarily mapped to each other while the MAGAN maintains the internal structure of the dataset. Similarly, in Figure 3c and 3d, there is the MNIST dataset with the first domain comprising 3's and 7's and the second domain being the first domain rotated $120°$. Without the correspondence loss, the subpopulations in the first domain are mapped arbitrarily to the subpopulations in the second domain, while the MAGAN maps original 3's to rotated 3's and original 7's to rotated 7's.

Figure 4 illustrates this on real biological data, where there are measurements from mass cytometry, consisting of individual cells with measured abundances of dozens of proteins. The two domains are separate experiments, and due to physical challenges replicating precise experimental conditions, in one domain, the abundance of a particular protein was systematically measured too low (*dropout*). Two subpopulations (*cell types*) in each experiment can be identified by looking at the values of CD45RA and CD45RO (naive T-cells and central memory T-cells) Capra et al. (1999). The dropout makes matching these cell types non-trivial, and a GAN without the correspondence loss incorrectly finds correspondence between naive T-cells in the first experiment and central memory T-cells in the second experiment (Figure 4a). The MAGAN correctly matches naive T-cells to naive T-cells and central memory T-cells to central memory T-cells. Crucially, with the MAGAN, accurately combining information from these two experiments is made possible.

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
