# OpenReview forum: "MAGAN: Aligning Biological Manifolds"
_ICLR.cc/2018/Workshop — Reject_

### Official Review · AnonReviewer3 · 2018-03-04
**MAGAN**

**Rating:** 3
**Confidence:** 5

**Review:**

The work describes how GAN combined with a correspondence loss can 'align' manifolds, where manifolds are low-dimensional data representations and where data can be obtained from different/same domains. The work touches upon an important problem of aligning the manifolds first rather than mere superimposing.

It is hard to evaluate what is the novel contribution in this work.
1. What is the functional form of the correspondence loss used? Without this equation, it is hard to understand why MAGAN really works. There are published papers that discuss aligning data using Least-squares or Variational divergences so this work is not the first of its kind.
2. Must your manifolds from different domains have same dimensions (cells or rows)? If so, how do you extract the same number of dimensions from both domains?
3. You might have one pair of data that is dense whilst the other pair data is sparse (due to dropouts etc). You could potentially end up with different number of clusters per domain. How does MAGAN now map manifolds?
4. How does MAGAN perform if you have overlapping clusters in either one or both manifolds?

---

### Official Review · AnonReviewer2 · 2018-03-08
**Interesting idea, but many important details are not presented clearly in the paper.**

**Rating:** 5
**Confidence:** 4

**Review:**

The authors proposed to use a GAN in order to align manifolds of two data domains in an unsupervised fashion, in hope that it gives natural pairing of data points across the domains. This has a lot of interesting applications in biology where it is very difficult and expensive to make various readouts jointly at the level of single cell, and the proposed method could potentially integrate readouts which are made independently.
The idea sounds interesting, but the paper is lacking a clear presentation of details such as the network architecture, and more importantly the "correspondence loss". The latter is specially important because it is where the alignment comes into play. Unfortunately this was not explained in the paper (e.g. Figure 2 was not reference at any point in the paper!).
In the experimental results, the authors showed that by introducing the correspondence loss, mapping by the trained network  becomes more biologically meaningful (at a very coarse level that at least the cell types match). I think to prove that the idea is useful, the evaluation should be done at finer levels too.

---

### Official Review · AnonReviewer1 · 2018-03-10
**hard to assess**

**Rating:** 5
**Confidence:** 3

**Review:**

This paper presents a method to align two manifolds such that related points in each measurement space are aligned together.

Cons:
The paper is hard to unpack. I advise the authors to re-write the first few paragraphs to clearly state a motivating example, a formal problem statement and a summary of the contributions of the paper. Right now it's too hard to even begin assessing the contribution. Finally, if the target application domain of this method is really computational biology, the paper will need a longer-form submission to explain 1) how precisely it handles batch effects 2) robustness to cases where one or more samples don't match across experiments.

---

### Decision · Program_Chairs · 2018-03-20
**ICLR 2018 Workshop Acceptance Decision**

**Decision:**

Reject

**Comment:**

Based on the reviews, this paper has not been accepted for presentation at the ICLR workshop. However, the conversation and updates can continue to appear here on OpenReview.